# Current and lagged associations of meteorological variables and *Aedes* mosquito indices with dengue incidence in the Philippines

**Estrella I. Cruz[1☯†], Ferdinand V. Salazar[1☯]\*, Ariza Minelle A. Aguila[1], Mary Vinessa Villaruel-Jagmis[1], Jennifer Ramos[1], Richard E. Paul[2]\***

**1** Department of Medical Entomology, Research Institute for Tropical Medicine, Filinvest Corporate City, Alabang, Muntinlupa City, Philippines, **2** Ecology and Emergence of Arthropod-borne Pathogens unit, Institut Pasteur, Université Paris-Cité, Centre National de Recherche Scientifique (CNRS) UMR 2000, Institut National de Recherche pour l'Agriculture, l'Alimentation et l'Environnement (INRAE) USC 1510, Paris, France

☯ These authors contributed equally to this work.
† Deceased.
\* fervillzar@yahoo.com (FVS); rpaul@pasteur.fr (REP)

**Data Availability Statement:** Data are freely available on Figshare private link: https://figshare.

## Abstract

### Background

Dengue is an increasing health burden that has spread throughout the tropics and sub-tropics. There is currently no effective vaccine and control is only possible through integrated vector management. Early warning systems (EWS) to alert potential dengue outbreaks are currently being explored but despite showing promise are yet to come to fruition. This study addresses the association of meteorological variables with both mosquito indices and dengue incidences and assesses the added value of additionally using mosquito indices for predicting dengue incidences.

### Methodology/Principal findings

Entomological surveys were carried out monthly for 14 months in six sites spread across three environmentally different cities of the Philippines. Meteorological and dengue data were acquired. Non-linear generalized additive models were fitted to test associations of the meteorological variables with both mosquito indices and dengue cases. Rain and the diurnal temperature range (DTR) contributed most to explaining the variation in both mosquito indices and number of dengue cases. DTR and minimum temperature also explained variation in dengue cases occurring one and two months later and may offer potentially useful variables for an EWS. The number of adult mosquitoes did associate with the number of dengue cases, but contributed no additional value to meteorological variables for explaining variation in dengue cases.

### Conclusions/Significance

The use of meteorological variables to predict future risk of dengue holds promise. The lack of added value of using mosquito indices confirms several previous studies and given the

com/s/0c5efc295eac6a3f09b5 and doi: 10.6084/m9.figshare.23978448.

**Funding:** Department of Science and Technology, Philippine Council for Health Research and Development to EIC and FVS. www.pchrd.dost.gov.ph Project entitled "Seasonal fluctuations of dengue vectors in selected endemic areas in the Philippines" under the MECO-TECO cooperation. French Agence Nationale de la Recherche (ANR), MO3 project, grant ANR-19-CE03-0004-01 to REP. The funders had no role in study design, data collection and analysis, decision to publish, or preparation of the manuscript.

**Competing interests:** The authors have declared that no competing interests exist.

onerous nature of obtaining such information, more effort should be placed on improving meteorological information at a finer scale to evaluate efficacy in early warning of dengue outbreaks.

## Author summary

Dengue is a widespread mosquito-borne disease. Mosquitoes are sensitive to temperature and rainfall and hence there have been efforts to identify such variables for predicting dengue outbreaks. Several mosquito indices are measured routinely by national surveillance systems, but which vary considerably in their success of predicting dengue outbreaks. This study explored the current and lagged associations of meteorological variables with mosquito indices and dengue incidence. Associations of mosquito indices with dengue were also explored. Rain and the diurnal temperature range (DTR) contributed most to explaining the variation in both mosquito indices and number of dengue cases. DTR and minimum temperature also explained variation in dengue cases occurring one and two months later. Mosquito indices did not provide any additional explanatory power for dengue incidences. Given the onerous nature of measuring mosquito indices, advanced warning systems might be improved using meteorological variables measured at finer scales than that traditionally available.

## Introduction

Dengue is a rapidly spreading mosquito-borne infectious disease caused by any of the four serotypes of the dengue virus (DENV 1–4). Despite the known underestimation of its real global burden [1], it is estimated that dengue incidence has increased 30-fold over the last few decades [2]. The disease is endemic in over 100 countries and more than 3.5 billion people are at risk of DENV infection [3,4]. Southeast Asia and the Western Pacific have historically been and are still among the most affected places [5–8]. The public health significance of dengue in the Philippines has continued since its initial discovery in 1954 to date [9]. Based on surveillance data in the Philippines for 2010–2014, Undurraga et al. estimated there were 794,255 annual dengue episodes [10], illustrating the high burden in the population.

Several environmental and socio-economic factors such as weather, urbanisation and globalisation have been associated with the spread of dengue [11–15] and this spread depends on the presence and abundance of the arthropod vector [16]. Moreover, in light of recent anthropogenic environmental impact, there has been a growing concern over global climate change as a potential factor increasing the risk of dengue through both the increased distribution of the mosquito vector species and the increased capacity for the vector population to transmit the virus (the vectorial capacity) [17–19].

DENV transmission is shaped by climatic conditions such as temperature and rainfall [20]. *Aedes* spp. reproductive and feeding behaviours, as well as viability of the species, depend, at least partly, on these environmental variables and thus have been widely studied. Mosquito abundance is partially regulated by rainfall by providing oviposition sites and triggering egg hatching [21]. Temperature influences the life span of the mosquito vector and has a direct effect on developmental and feeding rates [21–24]. In addition to this, DENV replication inside the vector also increases in warmer temperatures [23–25].

Having no fully effective vaccine to prevent infection, or drugs to treat the infection, vector control is still the main strategy to prevent transmission of DENV. Vector monitoring and surveillance is an evidence-based, analytical approach to better understand the mosquito vector population dynamics and virus transmission. Combined with a community-based strategy, which requires direct and immediate action in the community, vector surveillance offers the best defense against the vector and disease [26–28]. In urban settings, the major vector species is *Aedes aegypti*, which has adapted to the peridomestic environment, ovipositing in man-made artificial containers. Several *Aedes* indices, such as the House, Container, Breteau and Pupal indices (see Methods for definitions), have been proposed for monitoring strategies to predict risk of dengue transmission. However, to date there is little consensus on the appropriate threshold values to trigger a mosquito control response and their general utility to enable measures to be taken to avert an outbreak [29–33].

Identifying the factors that are implicated in DENV transmission and being able to forecast the onset of dengue outbreaks in endemic areas would provide the opportunity to be prepared and implement timely responses to decrease the burden of dengue in human populations. Efforts have been made to predict the incidence of infection using meteorological data [34–36]. However, our understanding remains poor over the relative contributions of meteorological variables in the context of urban settings where the microenvironment and the abundance of artificial oviposition sites impact mosquito abundance [18,19]. While other prevention and control measures for dengue are being developed, understanding the relationship between the risk of disease and meteorological factors in an urban setting for elaborating early warning systems (EWS) remains of key interest.

The aim of this study was to expand the current knowledge on the associations between different meteorological variables and both the *Aedes* indices and the incidence of dengue and to assess the extent to which the mosquito indices, which are laborious to perform, bring added value to meteorological variables for explaining variation in dengue incidences in three representative cities in the Philippines.

## Methods

### Ethics statement

This study was granted approval by the Institutional Review Board of the Research Institute for Tropical Medicine, the Philippines, approval number 2013–007.

### Study sites (Fig 1)

The study was carried out in six barangays, of which two were in each of the three study cities, Manila and Muntinlupa on the major island of Luzon and Puerto Princesa on Palawan island. In the highly densely populated area of Manila, two barangays were selected: Sampaloc, Manila (14˚ 36' 39.708"N 120˚ 59' 46.4496"E) covering 7.90 km$^2$ with a population of 388,305 and Tambunting, Santa Cruz District, Manila (14˚37'46.3"N 120˚59'01.5"E) covering 3.07 km$^2$ with a population of 126,735. In the more recently urbanized and less dense city of Muntinlupa, two barangays were selected: Cupang (14˚25'53.4″N 121˚2'55″E) covering 5.37 km$^2$ with a population of 57,013 inhabitants and Putatan (14˚23'54.12″N 121˚2'10.96″E) covering 6.75 km$^2$ with a population of 99,725. On the island of Palawan, which has a much lower degree of urbanization and population density, two urban barangays were chosen within Puerto Princesa, a coastal city covering 2,381 km$^2$ with a population of 307,079. The two selected barangays were: San Miguel (09˚44'39.48"N 118˚44'44.16"E), considered an urban barangay with a population of 21,157 and San Pedro (09˚45'19.44"N 118˚45'2.52"E), a neighboring urban barangay with a population of 25,909.

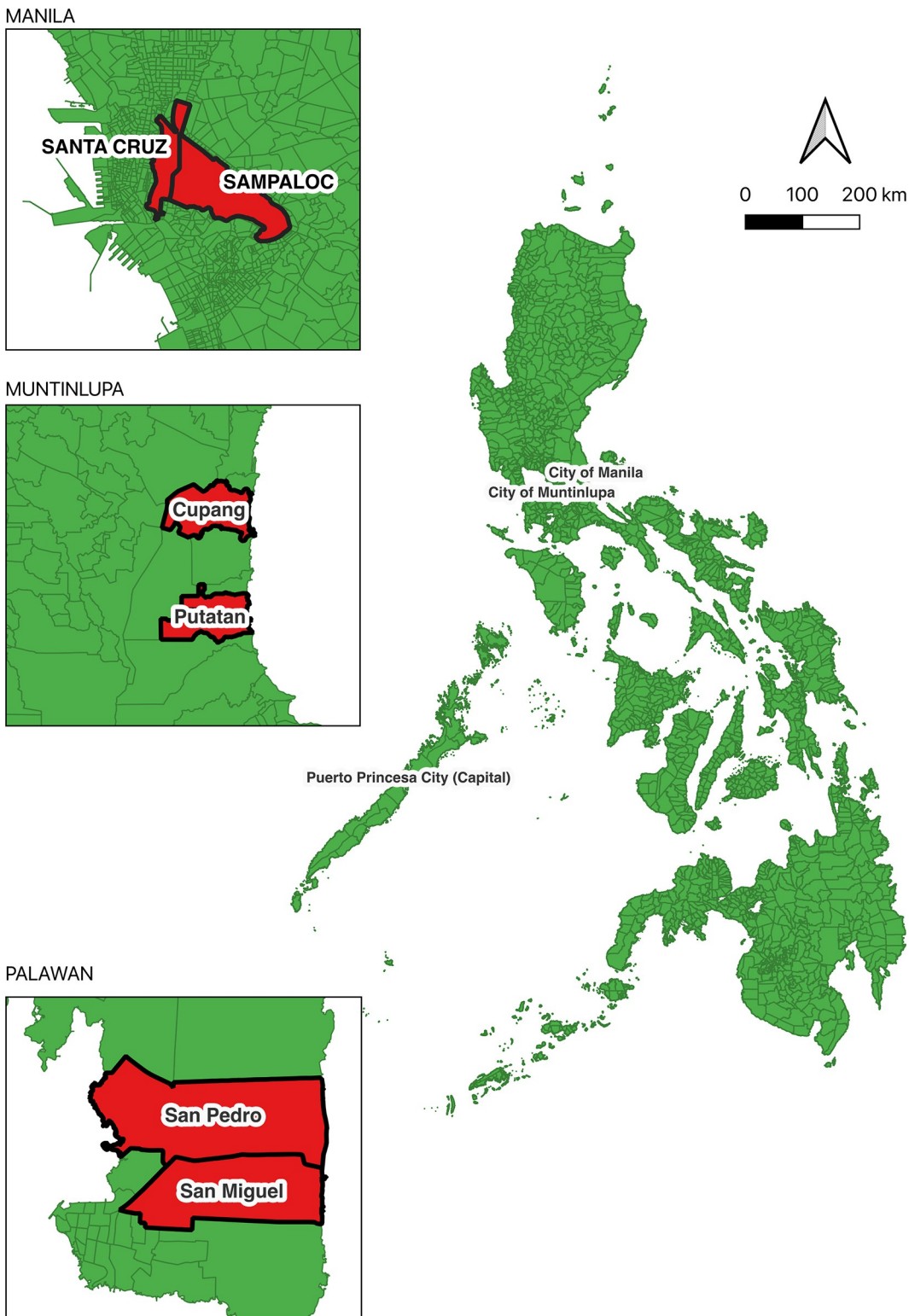

**Fig 1. Study Sites (Downloaded from https://data.humdata.org/dataset/cod-ab-phl?).**

### Entomological, meteorological and dengue case data

Entomological surveys were carried out monthly from November 2014 until December 2015, thus 14 months duration. Dengue case data were available monthly from October 2014 until December 2015. It subsequently transpired that the numbers of dengue cases during these two years were lower than average [37], despite there being an El Niño event during 2015–2016. Meteorological data were available on a daily basis, provided by the Philippine Atmospheric, Geophysical and Astronomical Services Administration (PAG-ASA) from meteorological stations closest to the areas under study. Meteorological data included rain, relative humidity and temperature. From these data the cumulative monthly rain (mm), the monthly mean daily rain (mm), the monthly mean daily Relative Humidity (%), the monthly mean daily minimum, maximum and mean temperatures (˚C) and the monthly mean daily Diurnal Temperature Range (DTR,˚C) were calculated.

Mosquito indices: Immature mosquito life stages surveillance was conducted in 100 households per selected barangay wherein all water-holding containers were inspected for presence of immature mosquito stages. Inspection was carried out within and immediately surrounding the house; i.e. in their yards/gardens. Containers inspected were majoritarily artificial, but did include organic waste (discarded coconut shells) and bamboo stumps of palm plant axials where present. Larvae and pupae found in the containers were all collected by pipetting. Specimens from each positive container were transferred into separate plastic bags for transport to the laboratory for rearing and identification. The number of water-holding containers (artificial or natural) with or without cover that were found indoors or outdoors, together with the number of containers found positive for larvae and/or pupae were recorded. The number of people who slept inside the house the previous night was also noted. The following indices were used: the Container Index (CI, percentage of water-filled containers positive for larvae or pupae), the House Index (HI, percentage of houses found positive for larvae or pupae), the Breteau index (BI, the number of containers positive divided by the number of houses visited), the Pupal Index (PI, the number of pupae divided by the number of houses visited x 100), the number of pupa per person (PPI, the number of pupae divided by the total population of the inspected households). In addition to these immature *Aedes* spp. mosquito stage indices, adult mosquitoes were captured in the houses using a sweep net each month. Adult mosquitoes were collected by circumnavigating the internal periphery of the house from the front door to the different rooms while continually moving the net in a figure of eight at 90˚ or at 180˚ targeting known resting places of adult mosquitoes. These areas include areas under the beds, hanging clothes, under the sink, comfort rooms, closets, dark cool rooms of the house, shoe racks, and outdoors such as vegetation, bushes, trees and plants. Trapped mosquitoes were transferred to Styrofoam cups with the use of sucking tubes. All collected mosquitoes were identified morphologically to species in the laboratory. In addition to the monthly adult *Aedes* spp. mosquito count, a cumulative two-monthly count was calculated.

### Statistical analyses

To test the association of the meteorological variables with the mosquito indices, a Generalized Additive model (GAM) with a spline function was fitted to account for any non-linearity in the relationships. For indices concerning percentages (i.e. House Index and Container Index), a logistic regression with a logit link function was fitted. For the indices concerning counts, a Poisson loglinear regression with a logarithmic link function was fitted.

Thus for the case of the Poisson log-linear models, the following regression equation was implemented:

Y (mosquito index or dengue cases)~Poisson (μt)

$$\log(\mu t) = \alpha + \beta 1(City)t + \beta 2(City.Barangay)t + s(\text{meteorological variable}, \lambda)t$$ where $\log(\mu t)$ is the logarithm of expected mosquito index N or dengue cases at time point t, $\alpha$ is the model intercept, $\beta$ are the beta coefficients and $s(\text{variable}, \lambda)$ is the natural cubic spline smoothing function.

First, all variables were fitted in a univariable analysis and those achieving a P-value less than 0.25 were fitted in the multivariable analysis. Barangay was nested within City and fitted as a fixed factor. The multivariable analysis proceeded by step-wise elimination of non-significant variables until a final adequate model containing only significant variables was achieved. Thus, for example, for the association of the seven meteorological variables with each mosquito index, there were eight models fitted (seven univariable and one multivariable). Because many of the variables were strongly correlated (i.e. the min, mean and max temperature variables), and thus led to collinearity and potentially spurious non-significance in the multivariable analysis, models were refitted with removal and replacement to identify which of such variables, when significant, were the most strongly associated (based on % variance explained). If more than one such variable was significant, then all were retained in the final model. To explore the delayed effect of the meteorological variables on mosquito indices, weekly time lags were generated considering *Aedes* mosquitoes adult life span (i.e. ~15–20 days) as well as the dengue transmission cycle (i.e. ~15 days), thus approximately a month. Based on previous significant meteorological findings [35,38], lags of 1–4 weeks were generated. Thus, for example, for lag week 1, we took the meteorological data from the week prior to current month plus the meteorological data from the following three weeks to generate a monthly value. For lag week 2, we took the meteorological data from 2 weeks prior to the current month and the subsequent 2 weeks and again generated a monthly value. By lag week 4, therefore, we used the meteorological data from the month preceding the mosquito collections. As for the same month analyses, we first conducted univariable and then multivariable analyses per lag week. Finally, we performed a multivariable analysis including all significant lag week variables. Therefore, overall, there were five "lag" time points (0, 1, 2, 3, 4 weeks), eight models per lag plus a combined analysis including all the four lags (lag wks 1–4) giving 41 models per mosquito index. The significance threshold P value of 0.05 was thus divided by 41, giving a Bonferroni P threshold of 0.0012.

For associations with dengue, we approached this differently because the time scale for a putative effect of meteorological or entomological variables on dengue cases was deemed to be longer for several reasons. Firstly, the expansion of the mosquito population because of meteorological effects will take time and thus any consequent effect on virus transmission will be accordingly delayed. Secondly, because of the extrinsic incubation period (EIP). The first blood meal is generally taken three days after emergence of the adult mosquito [21]. Therefore, assuming an EIP of 7–12 days, a minimum of 10–15 days would be required for a newly emerged mosquito to become infectious if its first blood meal was on an infected person [39]. This additional delay will thus generate a lag between the mosquito indices and the dengue incidence. Thirdly, the probability of a mosquito having an infected bloodmeal will be low at the start of the epidemic and increase with increasing dengue incidences. Thus, there will be some delay during the initial phases of the epidemic prior to the expansion of the viral population within the community. For these reasons we assessed the association of the variables on the current month's dengue cases and those occurring one and two months later. Thus, seven univariable and one multivariable analysis was performed for the association with meteorological variables. Seven mosquito indices were used for association with dengue cases: HI, CI, BI, PI, PPI, Adults and cumulated adults over two months. A Poisson log-linear regression was again fitted, including the natural log of the population of each barangay as an offset. When using the mosquito indices as explanatory variables, because the distribution of the data are

not normal, the data were transformed either by using an arcsine transformation for the percentage indices or by standardization for the count indices (standardization is akin to normalization and is calculated as the subtraction of the mean from each data point and division by the standard deviation). A dispersion parameter was estimated to account for any over-dispersion of the data in all analyses. Relative Risk from the Poisson regression was calculated as the exponentiated value of the model parameter estimates for the linear parameters. All analyses were conducted in Genstat version 22 [40].

## Results

### Description of city specific meteorological variables, dengue cases and mosquito catches

Rainfall showed distinct seasonality, with the dry season from January to April and the peak rainy season from July to October (S1 Fig). There was little variation among the three cities, with the monthly mean of the daily rainfall varying from 0.014–14.35 mm in Muntinlupa, 0–9.24 mm in Manila and 0–10.16 mm in Palawan. Relative humidity was consistently lower from February to June in all three cities (S2 Fig). Palawan varied the least over the year (73.7–82.5%) as compared to Manila (61.8–84.6%) or Muntinlupa (65.8–81.0%). Maximum temperatures reached a peak from April to June (Maximum 35°C in Manila, 34–35°C in Muntinlupa and 32–33°C in Palawan) (S3 Fig). Year round temperatures varied little with minimum temperatures oscillating around 24–27°C and mean temperatures around 25–31°C (S4 and S5 Figs). Temperatures varied little in the three cities but were generally less variable in Palawan as compared to Manila and Muntinlupa. The same was seen for the Diurnal Temperature Range with a peak from March to June (8–9°C) and higher values in Manila and Muntinlupa than Palawan (6–9°C vs. 6–8°C) (S6 Fig).

Over the study period, dengue cases were concentrated from July-December 2015. Overall, there were 343 and 369 in Tambunting and Sampaloc (Manila), respectively, 76 and 240 in Cupang and Putatan (Muntinlupa), respectively, and 188 and 110 in San Miguel and San Pedro (Palawan). Dengue Incidence rates (IR) per 10,000 individuals were initially low (0–5.2 / 10000) until July when an epidemic occurred, notably in San Miguel, Palawan, with IRs ranging from 9.0–28.4 / 10000 (Fig 2).

Overall, 10,856 immature stage mosquitoes were collected, 5,353 males and 5,503 females: 85.4% of were *Aedes aegypti*, 12.3% *Aedes albopictus* and 2.4% *Culex* or other genus spp. In the Luzon study sites (Manila and Muntinlupa), there were 3,661 immature stages collected; 96.8% were *Ae. aegypti*, 1.9% were *Ae. albopictus* and 1.4% *Culex* or other genus spp. In the two Palawan study sites, 7,195 immature stages were collected; 79.5% were *Ae. aegypti*, 17.5% *Ae. albopictus* and 2.9% *Culex* or other genus spp. The most productive containers for immature stages in Manila and Muntinlupa were jugs/pitchers, dish drains, gallons, pails and drums. In Palawan, in addition to these container types, other productive containers were tyres, coconut shells, wells and garbage cans.

For the adult mosquito catches (N = 1,386, 735 Male and 651 Female), 50.5% were *Ae. aegypti*, 0.2% *Ae. albopictus* and 49.3% *Culex* or other genus spp. The differences in the relative percentages of *Culex* and *Aedes* spp. for their immature *vs.* adult stages reflects the different oviposition site preferences of these genera and the fact that the mosquito immature indices are specifically designed for *Aedes* spp. In Palawan, where there was the most *Ae. albopictus*, *Ae. albopictus* larvae were only found alone (in the absence of *Ae. aegypti*) in containers on 16 occasions out of 572 positive containers. Thus, for the subsequent calculation of mosquito indices, *Ae. aegypti* and *Ae. albopictus* numbers were combined. However, because of the differing relative abundance of *Ae. albopictus* in Palawan, for the association of mosquito indices

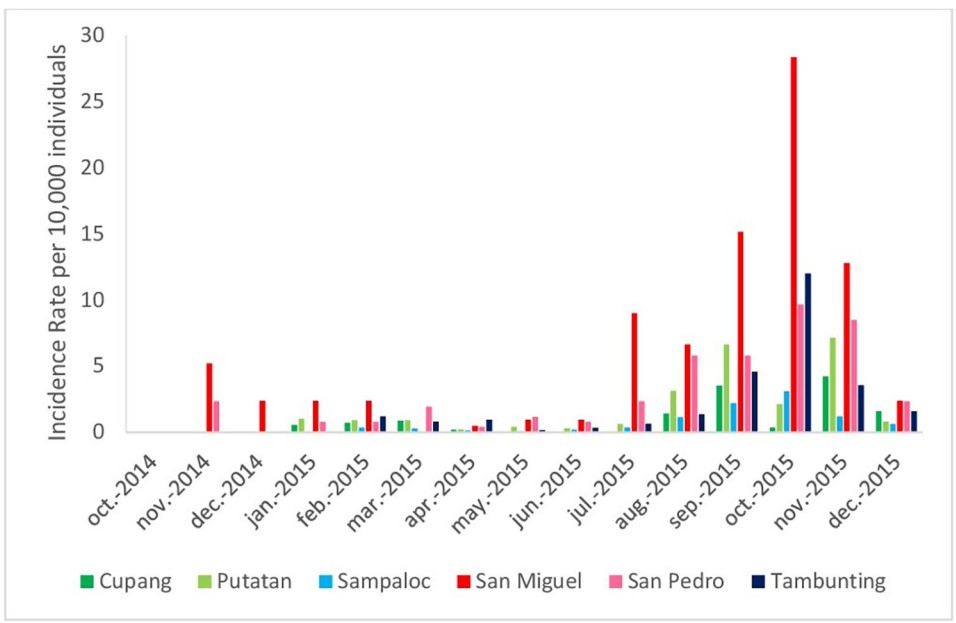

**Fig 2. Dengue Incidence Rate per 10,000 individuals in the six study sites.**

with meteorological variables we first analyzed all the sites together and then the two Palawan and four Luzon sites alone.

## Association of meteorological variables with mosquito indices

The House indices (HI) were very high, reaching 50% in some sites in the months following the onset of the rains in July (S1 and S7 Figs). Indices were highest in San Pedro and San Miguel in Palawan and Putatan in Muntinlupa, the latter being exceptionally high as compared to the other sites during the dry season, reaching 20%. Association analyses revealed that in addition to the variation explained by location (I.e. City and Barangay) (27.2% variance explained), meteorological variables explained 57.4% of the observed variation in HI (Table 1). The monthly mean of the Diurnal Temperature range exhibited an increase in HI to a peak at 6–6.5˚C before a subsequent decline (Fig 3). Cumulative monthly rainfall and mean daily rainfall showed the same relationship with HI, where HI increased gradually before reaching a plateau at ~150mm for cumulative monthly rain and ~5mm for mean daily rain per month (Fig 3).

Container indices (CI) were substantially lower, generally less than 5%, except reaching a maximum of 17% in San Pedro following the rains (S8 Fig). Once again, CIs were higher following the rains in San Pedro, Palawan and during the dry season in Putatan, Muntinlupa. The same relationships were observed with the meteorological variables as for HIs, explaining 50.6% of the variation in CIs (Table 1 and S9 Fig for the relationship with DTR).

Breteau indices in the two sites in Manila never exceeded a BI> = 5 (S10 Fig). In Cupang, Muntinlupa, BI> 5 occurred on three occasions but Putatan was consistently higher than 5 and again showed aberrantly high values (10–23) during the dry months. Values in the two sites in Palawan were consistently very high throughout the year except for the dry season months (February to June); values reached over 100 in San Pedro and 9–23 in San Miguel during the dengue epidemic period from July onwards in 2015. The same associations and relationships with meteorological variables observed for HI and CI were observed, explaining 63% of the variation in BI (Table 1).

**Table 1. Association of meteorological variables with mosquito indices collected in the same month, in the multivariable analyses.**

| | | P value | variance explained (%) |
|---|---|---|---|
| House Index | Cumulative Rain (mm) | < .001 | 13.37 |
| | Mean DTR (˚C) | < .001 | 20.71 |
| | Mean Rain (mm) | < .001 | 15.10 |
| | Mean RH (%) | < .001 | 5.01 |
| | Min. Temp. (˚C) | *0.01* | 3.22 |
| Container Index | Cumulative Rain (mm) | < .001 | 15.25 |
| | Mean DTR (˚C) | < .001 | 20.19 |
| | Mean Rain (mm) | < .001 | 6.45 |
| | Mean RH (%) | 0.001 | 5.39 |
| | Min. Temp. (˚C) | *0.016* | 3.34 |
| Breteau Index | Cumulative Rain (mm) | < .001 | 14.94 |
| | Mean DTR (˚C) | < .001 | 25.16 |
| | Mean Rain (mm) | < .001 | 14.57 |
| | Mean RH (%) | 0.001 | 4.14 |
| | Min. Temp. (˚C) | *0.002* | 3.82 |
| Pupal Index | Cumulative Rain (mm) | *0.009* | 6.90 |
| | Mean DTR (˚C) | < .001 | 17.38 |
| | Mean Rain (mm) | *0.009* | 6.85 |
| | Mean RH (%) | *0.007* | 7.30 |
| | Min. Temp. (˚C) | *0.011* | 6.61 |
| | Max. Temp. (˚C) | *0.043* | 3.58 |
| Pupa per person | Cumulative Rain (mm) | *0.007* | 6.94 |
| | Mean DTR (˚C) | < .001 | 18.03 |
| | Mean Rain (mm) | *0.013* | 6.14 |
| | Mean RH (%) | *0.006* | 7.19 |
| | Min. Temp. (˚C) | *0.01* | 6.44 |
| | Max. Temp. (˚C) | *0.033* | 3.78 |
| Adults | Cumulative Rain (mm) | < .001 | 36.26 |
| | Mean DTR (˚C) | *0.006* | 4.76 |
| | Mean Rain (mm) | *0.006* | 4.82 |
| | Mean RH (%) | < .001 | 7.22 |
| | Min. Temp. (˚C) | < .001 | 14.42 |
| | Max. Temp. (˚C) | *0.003* | 4.52 |

P values in italics are those above the Bonferroni corrected P threshold for multiple tests (P = 0.0012, See Methods).

The Pupal index was highly variable ranging from 0 to over 100% (S11 Fig); the Pupa per person indices were also highly variable, ranging from 0 to 0.25) (S12 Fig). With the exception of Putatan, which showed the aberrantly high pupal indices in the dry season months, PIs and PPIs were generally zero during the dry season and then rapidly increased in the months following the rains. The same meteorological associations were observed, albeit generally weaker than those observed for the HI, CI and BI. However, these variables still explained 49% of the observed variation in PI and PPI (Table 1).

In contrast to these immature stage indices, adult *Aedes spp.* were less abundant in the two Palawan sites and most abundant in Putatan, especially during the dry season (S13 Fig). The relationship of the meteorological variables with the number of adult *Aedes* spp. mosquitoes

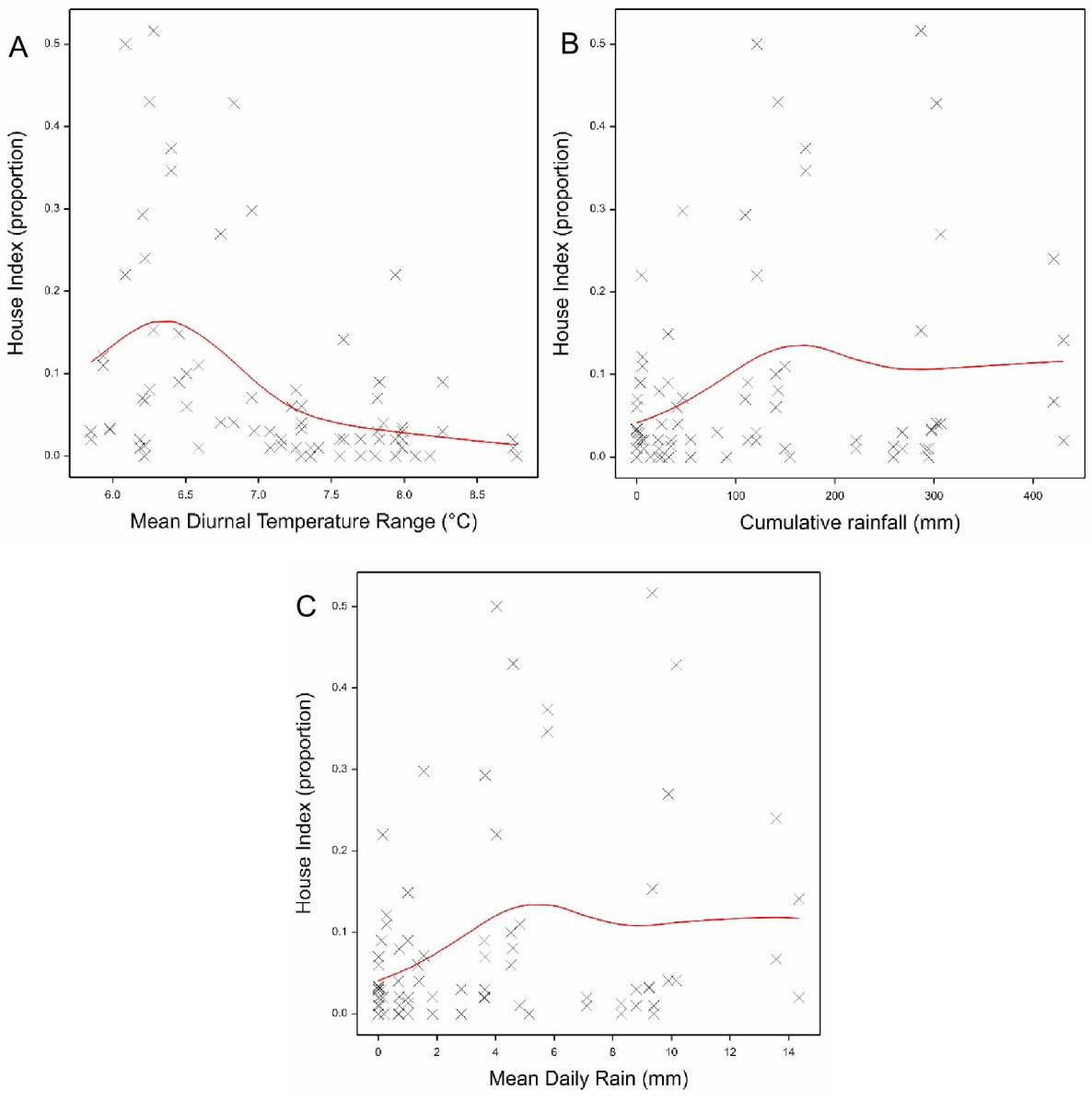

**Fig 3.** The fitted logistic regression of the House Index (here shown as a proportion) against (A) the Diurnal Temperature range, (B) cumulative rain and (C) mean rain. The red lines show the fitted model in the GAM.

also differed (Table 1). There was a strong association with cumulative rain, with progressively increasing numbers of *Aedes* spp. adults caught with increasing rain (Fig 4). There was a distinct non-linear relationship with minimum temperature, with a peak at 25–26°C (Fig 4). The strength of the relationships with the other meteorological variables was much weaker, but combined overall explained 72% of the observed variation.

Analysis of the two sites on Palawan separately from the Luzon study sites revealed a similar series of relationships with the meteorological variables. For HI, CI and Breteau, cumulative rain and DTR were significantly associated (cumul. Rain: % variance explained 43.8%, 39.8% and 46.9% for HI, CI and Breteau respectively; DTR explained 13.1%, 14.2% and 17.3% of the variance for HI, CI and Breteau respectively). For the Pupal Index, cumulative rain and

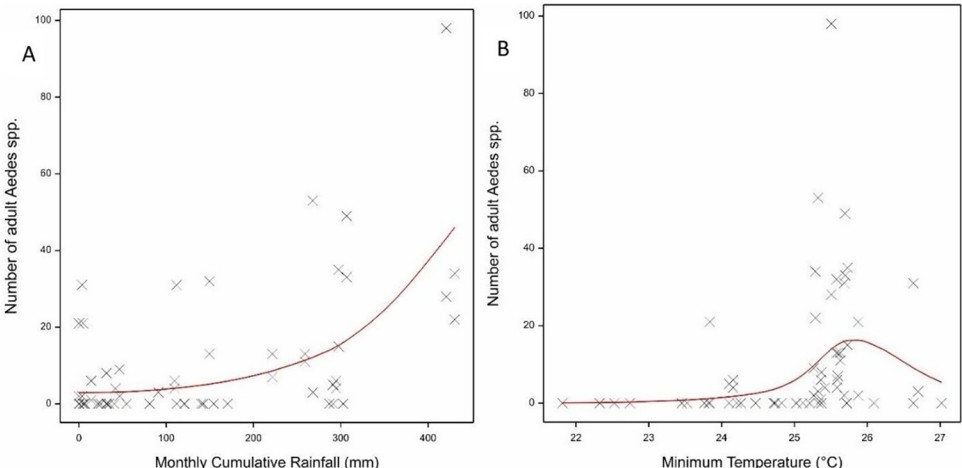

**Fig 4.** The fitted loglinear regression of the number of adult *Aedes* spp. mosquitoes against (A) the monthly cumulative rain and (B) the monthly mean minimum temperature. The red line shows the fitted model in the GAM.

Relative Humidity explained 28.4% and 31.0% of the variance. For the PPI, cumulative rain, DTR and Relative Humidity explained 26.2%, 17.5% and 20.4% of the variance. Finally, for the adults, cumulative rain and DTR were significantly associated, explaining 44.3% and 27.2% of the variance.

## Association of meteorological variables with lagged mosquito indices

The association analyses were then repeated for lag periods of one to four weeks as indicated in the methods. Monthly values of the meteorological variables were thus calculated for the weeks preceding the mosquito index counts. With the exception of the Pupal Index, lagged associations explained less of the observed variation in the mosquito indices (S1 Table). For the Pupal Index, variables lagged by two weeks explained 52.9% of the variation as compared 48.6% for the unlagged associations. In both cases there was a notable decrease in PI with increasing DTR (Fig 5) and a significant increase in PI with increasing cumulative rain (Fig 5).

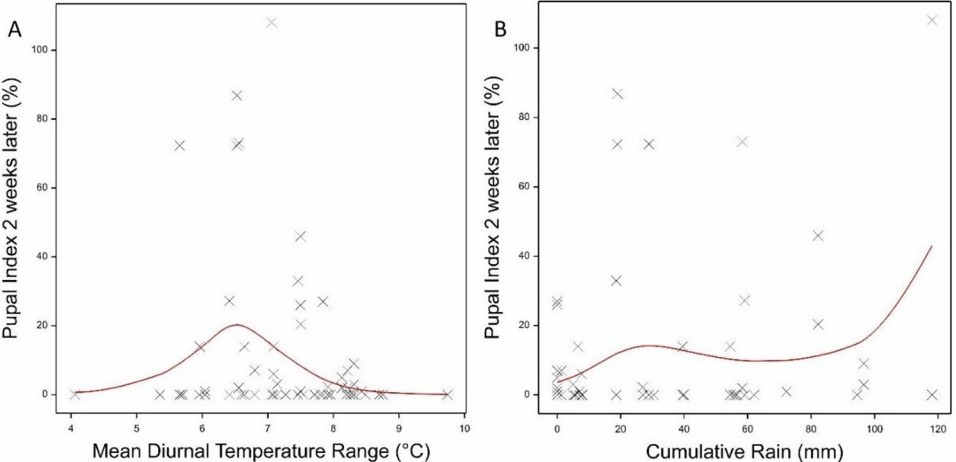

**Fig 5.** The fitted loglinear regression of the lagged 2 week Pupal Index against (A) the monthly mean Diurnal Temperature range (˚C) and (B) the monthly cumulative rain (mm). The red line shows the fitted model in the GAM.

Performing a combined analysis of all the significant variables identified in the individual four week lag times did not improve the model fit (% variance explained Current month vs lagged fits: HI: 57% *vs*. 25%; CI: 51% *vs*. 40%: BI: 63% *vs*.57%; PI: 49% *vs*. 40%; PPI: 49% *vs*. 36%; Adults: 72% *vs*.60%)(S1 Table).

### Association of meteorological variables with dengue cases

There were several associations of meteorological variables with the number of dengue cases during the same month (Table 2). There was a gradual increase in the number of cases with increasing cumulative rain, mean minimum temperature and DTR (Fig 6). Dengue cases peaked at a minimum temperature of 25–26˚C and a DTR of 6.5–7˚C. Overall the meteorological variables explained 60.4% in the variation of observed cases. Assessing the association with dengue cases the following one or two months explained less of the variation (38.9% and 39.3% respectively) and the rain variables became non-significant. However, the associations with DTR and especially minimum temperature remained significant and followed the same form as for the current month's dengue cases. Shown in Fig 7 are the model fits for mean daily minimum temperatures and dengue cases one and two months later.

Analyzing the two Palawan sites independently of the four Luzon sites revealed a much improved model fit, with cumulative rain, DTR and Relative Humidity explaining 71.3% of the variation in the number of dengue cases in the same month. Moreover, cumulative rain, DTR and mean maximum temperatures explained 73.9% and 71.7% of the variation in cases one and two months later respectively. The overall model fit for the association of the meteorological variables with the number of dengue cases two months later is shown in Fig 8 for each of the two Palawan sites. 10.2% of the variation was accounted for by the site (San Miguel vs. San Pedro). Likewise, when analyzing the four sites on Luzon independently of Palawan gave a much improved model fit, explaining 69.5%, 48.1% and 77.9% of the variation in dengue cases the same month and one and two months later. Minimum temperatures and DTR contributed the most. The forms of the associations of the individual meteorological variables with dengue cases was as observed in the combined site analysis (e.g. Fig 7 for minimum temperatures). The overall model fit for the association of the meteorological variables with the number of dengue cases two months later is shown in Fig 9 for each of the four Luzon sites. However, when analyzing Muntinlupa and Manila separately, there was no improved model fit from when analyzing all four Luzon sites together.

**Table 2. Association of meteorological variables with dengue cases in the multivariable analysis.**

|  | Variable | P value | variance explained (%) |
|---|---|---|---|
| Cases same month | Cumulative Rain (mm) | < .001 | 15.02 |
|  | Mean DTR (˚C) | 0.001 | 8.36 |
|  | Mean Rain (mm) | < .001 | 15.32 |
|  | Mean RH (%) | *0.01* | 5.75 |
|  | Min. Temp. (˚C) | < .001 | 11.54 |
|  | Max. Temp. (˚C) | *0.012* | 4.39 |
| Cases 1 month later | Mean DTR (˚C) | 0.002 | 10.90 |
|  | Mean RH (%) | *0.018* | 7.32 |
|  | Min. Temp. (˚C) | < .001 | 20.63 |
| Cases 2 months later | Mean DTR (˚C) | 0.001 | 11.38 |
|  | Mean RH (%) | 0.003 | 9.68 |
|  | Min. Temp. (˚C) | < .001 | 18.20 |

8 models per analysis. Bonferroni P value threshold P = 0.0063. P values in italics are those above this threshold.

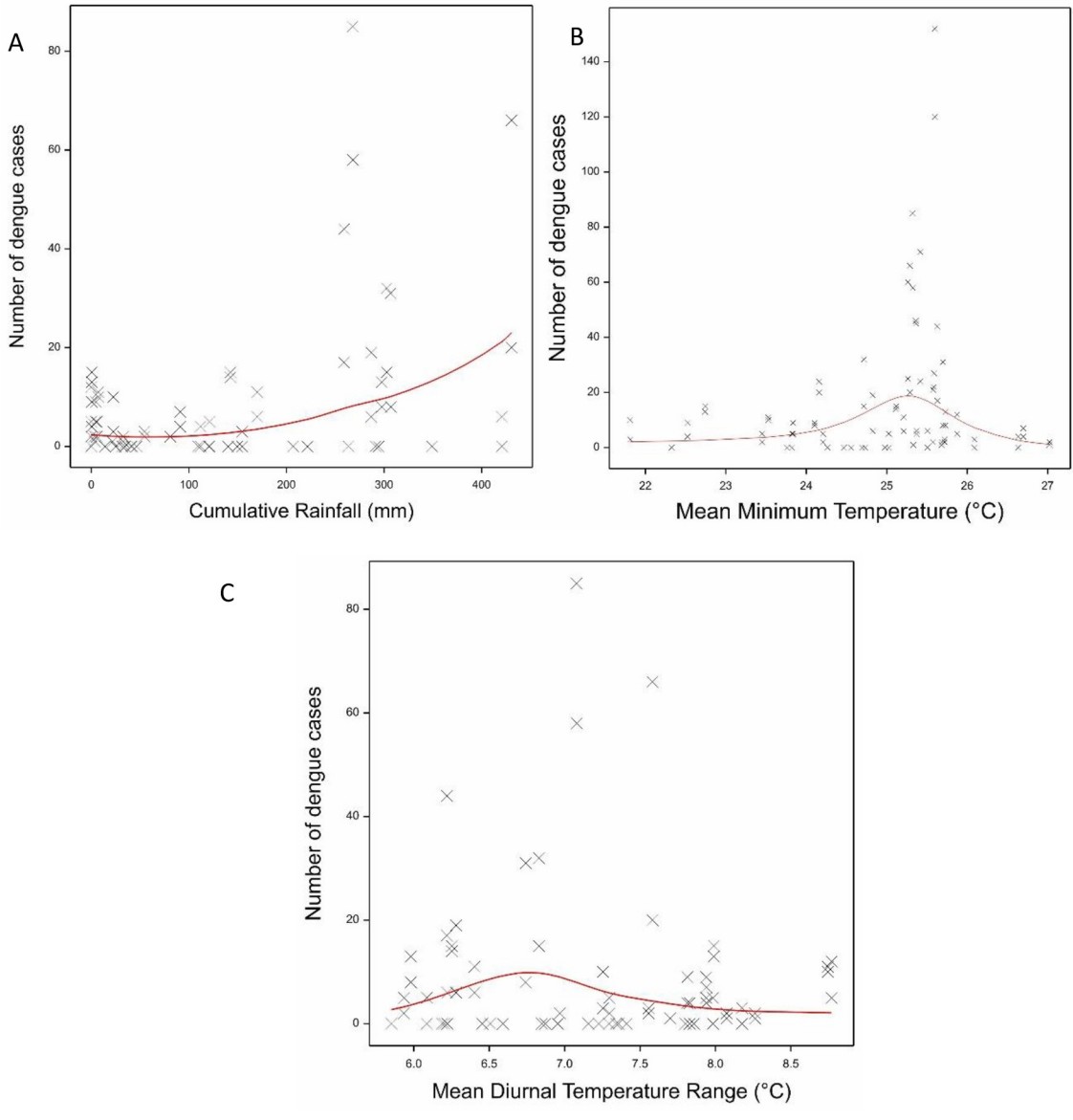

**Fig 6.** Fitted model of the association between (A) cumulative rain, (B) Mean Minimum Temperature and (C) DTR with dengue cases in the same month. Shown are the observed data (crosses) and the fitted model (red line).

### Association of mosquito indices with dengue cases

Whilst Container, House and Breteau indices and cumulative adult mosquitoes (over 2 months) were all associated with dengue cases in the same month in the univariable analyses, only the latter was positively associated with dengue cases in the final multivariable analysis (P = 0.001), explaining 4.79% of the variation in the number of dengue cases (Fig 10). There was an increased Relative Risk of 1.52 (95%CI 1.18–1.95) for every increase in one mosquito in the standardized mosquito number, equivalent to a Relative Risk of 1.02 (95% CI 1.01–1.03) for every increase in one mosquito (unstandardized). This positive association was lost when combined with the meteorological variables. There were no associations of mosquito indices with dengue cases the following one or two months.

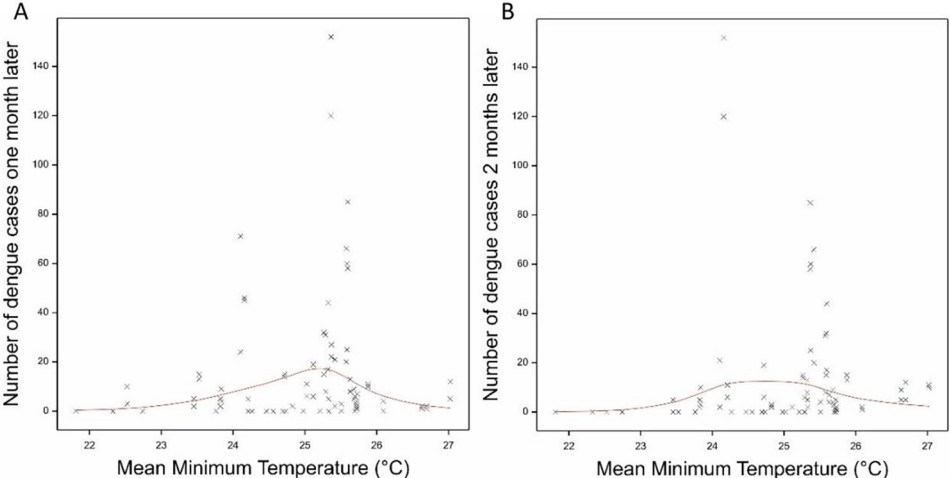

**Fig 7.** The fitted model association between monthly mean daily minimum temperatures and the number of dengue cases (A) one month later and (B) two months later. Shown are the observed data (crosses) and the fitted model (red line).

## Discussion

Meteorological conditions can influence the spread of dengue through their impact on the vector's life cycle and ability to transmit the virus. Because of this and the relative ease with which such data can be collected and collated, the use of meteorological variables for EWS of dengue outbreaks has been explored [34–36,41]. This study aimed to expand the knowledge on the associations and their time lags between different meteorological variables and mosquito indices on the one hand and dengue incidence on the other. This study also aimed to assess the added value of incorporating the mosquito indices in explaining variation in dengue incidence.

There were several non-linear associations between meteorological variables and the immature mosquito indices, namely HI, CI, BI, PI and PPI. The most pertinent variables were those associated with rain (whether cumulative or mean daily rain over the month) and the DTR. These variables explained 49–63% of the variation in the immature indices. During a longer time series study in Sri Lanka, cumulative rain greater than 200mm during the same month was associated with increased Breteau Indices [41]. The rain variables showed a distinct plateau, with increasing values of indices up a certain level of precipitation but not beyond. Heavy rain has been suggested to result in a decrease in immature mosquito indices by flushing out the immature stages, thus decreasing its population and ability to transmit the disease [35,42]. This study found no evidence of a decrease as might be expected if excess rain led to flushing of the oviposition sites. The plateauing out of the indices with increasing rain might reflect a saturation in the number of available water-filled oviposition sites. Although the number of potential containers increases with population density, a modelling study on the relationship between human and mosquito densities has previously suggested that water container number likely does not keep increasing after a given number of humans is reached [43]. Larval and pupal developmental rates increase with temperature and an increase in larval indices with temperature (>31.5˚C) has previously been observed at monthly lag times [41]. Interestingly, there were significant lagged associations with temperature variables and larval indices at lag times in this study, even if the meteorological associations gave an overall better fit with the current month's larval indices. The non-linear relationship of immature indices with DTR is a

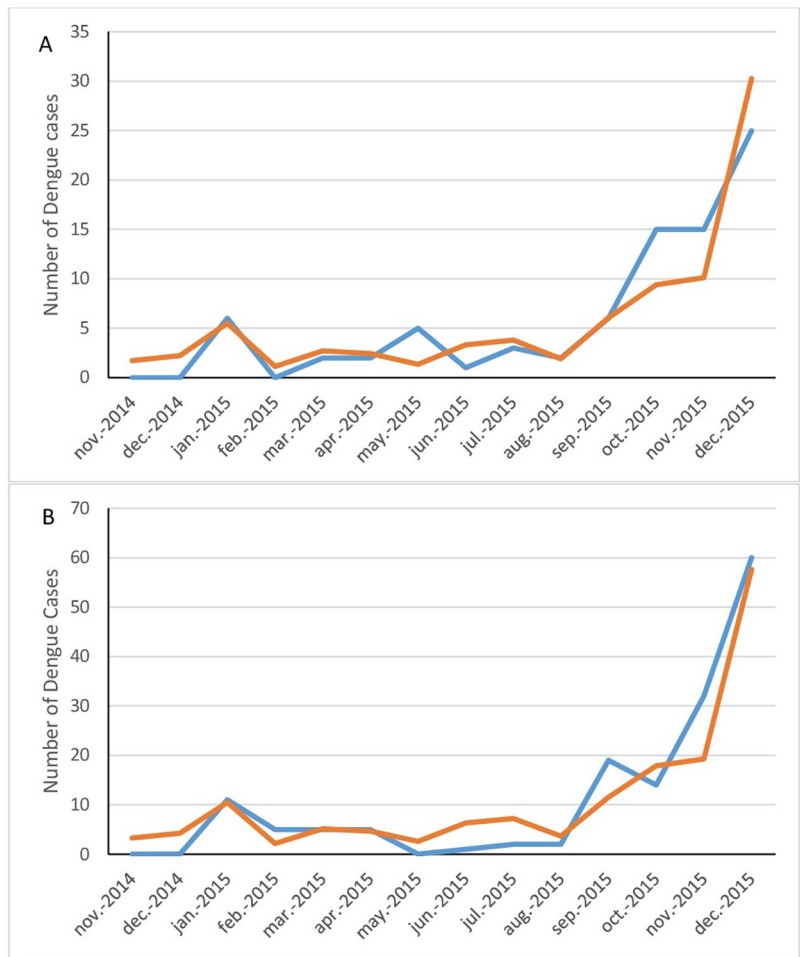

**Fig 8. The fitted model association of meteorological variables with the observed number of dengue cases two months later in the two study sites on Palawan.** (A) San Pedro, (B) San Miguel. Observed values shown by the blue line and fitted values by the orange line.

well-recognised phenomenon and developmental rates have been shown to decrease with increasing DTR. [44,45]. Furthermore, Carrington et al. found that both small and large DTRs can affect the population dynamics [46].

In contrast to the immature indices, the adult mosquito numbers were most strongly influenced by minimum temperatures and cumulative rain, with moderate effects of the other meteorological variables. No plateau in adult mosquito numbers was observed with increasing cumulative rain, but which may reflect the weak correlation between immature and adult numbers. The association with minimum temperature most likely reflects the more general effects of temperature on adult longevity with peak survival rates occurring round 27°C and increased mortality rates occurring above 32°C [47–49]. The temperature variables are all highly correlated and thus the minimum temperature *per se* may not be the most biologically important.

The absence of any improved model fitting when using mosquito indices lagged by one to four weeks after the meteorological variables suggests that the current month's weather is that impacting the current month's mosquito numbers. It should be noted, however, that although there was not an improved model fit, the meteorological variables in the lagged analyses did

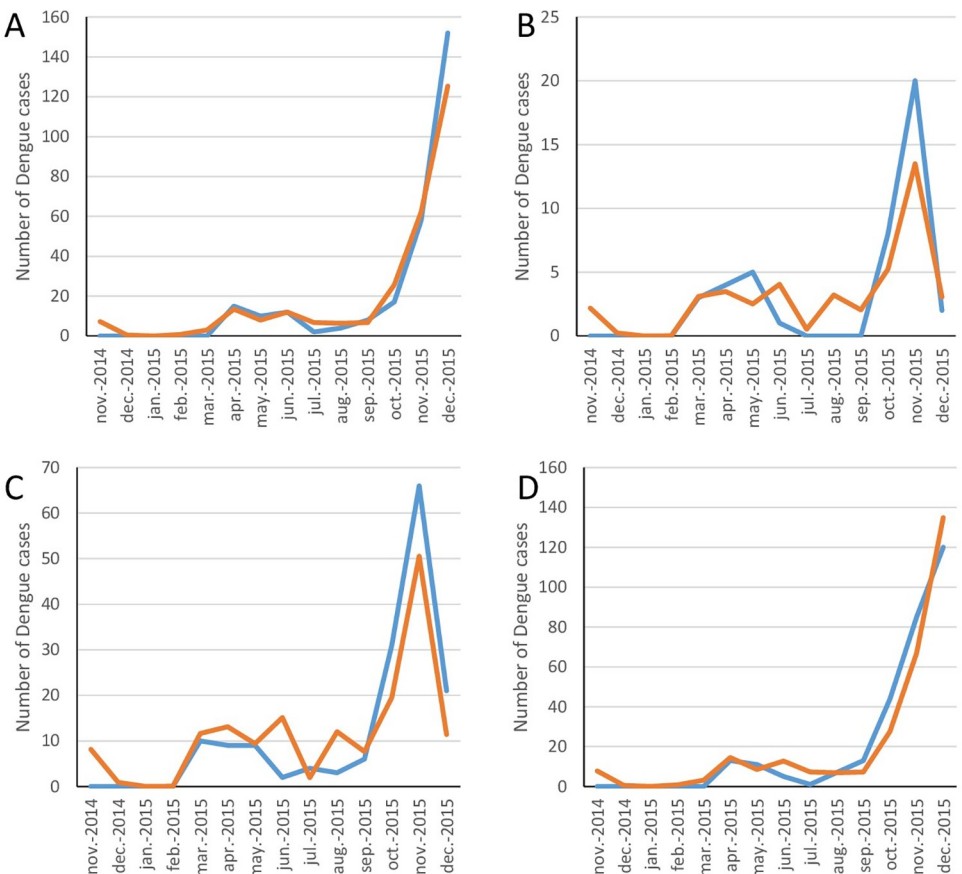

**Fig 9. The fitted model association of meteorological variables with the observed number of dengue cases two months later in Luzon.** (A) Cupang, (B) Putatan, (C) Sampaloc, (D) Tambunting. Observed values shown by the blue line and fitted values by the orange line.

explain a substantial amount of the observed variation, ranging up to 64% for a four week lag with adult mosquitoes (as compared to 72% for the same month's weather). Moreover, when we analysed the two islands separately, the lagged meteorological associations gave as good as, or even a better fit than the variables from the current month. This relatively strong association might be of value for an advanced warning index for an increase in mosquito densities. In addition, the absence of improved lag effects may also reflect the relatively low variation in the meteorological variables over such a short time period and a longer time frame might be more informative, especially with respect to the adult population density following the dry season.

Similarly to previous studies, most meteorological variables had a non-linear association with the incidence of dengue [35,38,50]. Temperature fluctuations have an impact on the mosquito's life span, development, reproduction rates, and feeding frequency, as well as the speed of virus replication and extrinsic incubation rate [21,51]. Low temperatures have been associated with a decreased vector capacity, and here we clearly observed an accelerating risk of dengue above a minimum temperature of ~22˚C. High temperatures have also been found to be associated with decreased risk of dengue, generating a non-linear pattern [24,35,38,52]. In an in-depth analysis, Mordecai et al. 2019 derived trait thermal performance curves from experimental data and found the thermal optimum for *Ae. aegypti*-vectored dengue transmission occurs at 29.1˚C (and thermal maxima and minima of 34.5˚C and 17.8˚C, respectively). For *Ae. albopictus* these values were 26.4˚C (optimum) and 31.4˚C and 16.2˚C for the maximum

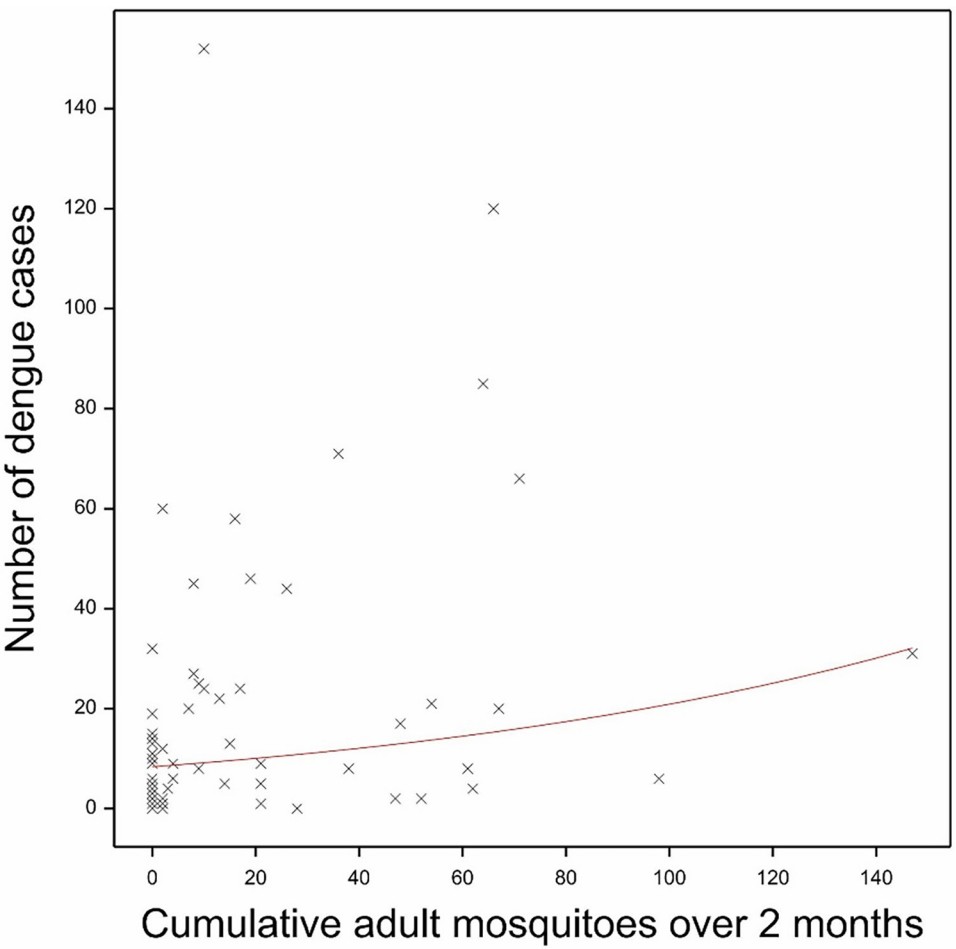

**Fig 10. The fitted loglinear regression of the standardized cumulative adult mosquitoes against the number of dengue cases the same month.** The red line shows the fitted model in the GAM.

and minimum temperatures [24]. This thermal maximum is also observed in the present study, with a trend for decreased dengue incidence at temperatures beyond 33˚C. A suggestive decrease in dengue risk was also observed beyond 34˚C in Guangzhou [53]. Previous estimates of threshold temperature values for increased risk of dengue incidence, all from Guangzhou, China have identified values of 18–23˚C and 22–32˚C for minimum and maximum temperature thresholds [53–55]. The values we found fall within the same range.

Lower diurnal temperature ranges have been associated with a higher spread of the dengue [22]. Moreover, less temperature variation during the day can also coincide with an optimal temperature mean for dengue transmission. A modelling approach found that decreased temperature variation around the predicted optimal temperature (~29˚C) increases the chances of viral transmission [23]. This negative association between temperature variation and dengue incidence was observed here and found only to lead to reduced risk at DTR above 6.5˚C. Below this DTR, mean temperatures ranged from 27–29˚C, suggesting an optimal combination of temperatures for transmission. It should, however, be noted that increasing DTR has been predicted to increase vectorial capacity at lower temperatures (~14˚C), highlighting the complex temperature effects on viral transmission [23].

Rainfall can increase dengue transmission, generating more abundant oviposition sites and maintaining a sufficient relative humidity for adult mosquito survival [21]. A positive

association with cumulative rainfall was observed here, with incidence increasing with as little as 100mm rainfall in a month. The similarity in the shape of the relationship of rain with adult mosquitoes and with dengue cases would suggest that this latter is occurring through the increase in adult mosquito numbers during that same month. This is supported by the observation of a positive association of cumulated adult mosquito numbers and dengue incidence.

When analysing the study sites on the island of Palawan separately from those on Luzon led to vastly improved model fit, especially for dengue cases occurring two months later. The same occurred when analysing the Luzon sites separately from the Palawan sites, but no added value was observed when analysing each city within Luzon separately. It is tempting to hypothesise that the weather variables are differentially affecting the two *Aedes* species that have differing relative abundances on the two islands with knock-on effects for explaining variation in dengue incidence. That the significantly associated meteorological variables with dengue cases in Palawan remained over one to two months lag time is consistent with previous findings and potentially useful for any advanced warning strategy [35,38]. Such long delayed effects are biologically plausible, reflecting the delay between amplification of the mosquito population, initial spread and subsequent expansion of the viral population, not to mention the reporting delays. DENV has a lifecycle taking a minimum of 15 days, including a ~10 day extrinsic incubation period within the mosquito following an infective bloodmeal and then a 4–10 day intrinsic incubation period following an infectious bite on a human. Two months lag would thus correspond to a maximum of three generations of viral transmission even with an adequate mosquito population density. Furthermore, given that the majority of DENV infections are inapparent, the consequences of expanding viral circulation identified through clinical cases would take longer [56].

This study identified two meteorological variables, DTR and minimum temperature, which were found to be associated with dengue cases one and two months later. Previous analyses found, amongst other meteorological variables, mean or minimum temperature to be associated with dengue cases occurring one month, 2–3 months and 1–12 weeks later according to the study [35,36,57]. This consistent association of temperature being associated with increasing dengue incidences in geographically different settings (Thailand, Brazil, Mexico, Barbados) is very promising for its use in a EWS [35,36,57]. The clear associations of meteorological variables and the mosquito indices is encouraging and yet it seems that the link between such indices and dengue incidence remains difficult to detect. Only adult mosquitoes showed any positive association with dengue incidence, but far weaker than when using meteorological variables. Previous work examining the association between mosquito indices and dengue cases did reveal associations with BI, CI and HI at lags of 1–2 months, but did not consider simultaneously the effect of the meteorological variables [58]. Despite recommendations from stakeholders testing the most elaborate EWS to date, EWARS, to include mosquito indices, our study suggests that the indices bring no added value [35].

There are several limitations to this study. Firstly, we only had 14 months dengue case data upon which to base our analyses, but which was carried out in six sites on two environmentally differing islands. Secondly, even though we had meteorological data from each of the cities, we had no more specific information at the scale of each site and thus could not include a spatial element in the analyses. Intra-annual spatial variation in incidence was observed, but may more likely reflect the known clustered nature of dengue outbreaks and the impact of herd immunity [59,60]. Whilst meteorological variables likely vary at very local scales, disentangling the relative roles of immunity, clustering and meteorology will require fine-scale measures of serology, meteorology and case geolocalisation. However, efforts to calibrate local microclimate with satellite data would be a major step in addressing the real associations of meteorological variables with mosquito indices.

In conclusion, there have been increasing efforts to establish early warning and response systems using meteorological information, but which although promising have often proven country-specific [34,35,57]. Our study does, however, suggest that there are globally consistent meteorological variables providing some predictive power at lead times of 1–2 months. It also reaffirms the lack of explanatory power of mosquito indices and hence questions the utility of investing the substantial effort necessary for collecting such indices. The absence of added value of mosquito indices may be because the meteorological variables strongly influence the mosquito bionomics and vectorial capacity and thereby better capture the associations with dengue incidence than mosquito indices can. It is possible, however, that at finer spatial scales mosquito indices may bring additional predictive value, being affected by the microclimate that the meteorological stations cannot capture. Further studies focussing on the more local scale with respect to both microclimate and mosquito indices might be a promising route of investigation.

## Supporting information

**S1 Fig. Mean Daily Rain (mm) by month in each of the three cities.** Shown are means and standard errors of the mean.
(TIF)

**S2 Fig. Mean Daily Relative Humidity (%) by month in each of the three cities.** Shown are means and standard errors of the mean.
(JPG)

**S3 Fig. Mean Daily Maximum Temperature (˚C) by month in each of the three cities.** Shown are means and standard errors of the mean.
(JPG)

**S4 Fig. Mean Daily Minimum Temperature (˚C) by month in each of the three cities.** Shown are means and standard errors of the mean.
(JPG)

**S5 Fig. Mean Daily Mean Temperature (˚C) by month in each of the three cities.** Shown are means and standard errors of the mean.
(JPG)

**S6 Fig. Mean Daily Diurnal Temperature Range (˚C) by month in each of the three cities.** Shown are means and standard errors of the mean.
(JPG)

**S7 Fig. House Indices per month in the six study sites.** Shown are % and 95% Confidence Intervals.
(TIF)

**S8 Fig. Container Indices per month in the six study sites.** Shown are % and 95% Confidence Intervals.
(TIF)

**S9 Fig. Fitted logistic regression model of Container Index (here given as a proportion) against DTR during the month.** Red line shows the model output for the association of DTR with CI and the actual data are crosses.
(JPG)

**S10 Fig. Breteau Indices per month in the six study sites.**
(TIF)

**S11 Fig. Pupal Indices per month in the six study sites.**
(TIF)

**S12 Fig. Pupa per Person Indices per month in the six study sites.**
(TIF)

**S13 Fig. Number of adult Aedes spp. per month in the six study sites.**
(TIF)

**S1 Table. Association of meteorological variables with lagged mosquito indices.** For each index we do 4 (lag weeks) x7 (meteorological variables) univariable analyses plus 1 multi per lag week and one combining all the lags together so 33. Bonferroni correction P value = 0.0015. P values in italics are those above this P value threshold.
(DOCX)

## Acknowledgments

We are grateful to the City Health Office and Officers from the Cities of Manila, Muntinlupa, and Puerto Princesa, Palawan for allowing us to conduct the study in their barangays.

## Author Contributions

**Conceptualization:** Estrella I. Cruz, Ferdinand V. Salazar.

**Data curation:** Ariza Minelle A. Aguila, Mary Vinessa Villaruel-Jagmis, Jennifer Ramos.

**Formal analysis:** Richard E. Paul.

**Funding acquisition:** Estrella I. Cruz.

**Investigation:** Estrella I. Cruz, Ferdinand V. Salazar, Ariza Minelle A. Aguila, Mary Vinessa Villaruel-Jagmis, Jennifer Ramos.

**Methodology:** Estrella I. Cruz, Ferdinand V. Salazar, Ariza Minelle A. Aguila, Jennifer Ramos.

**Supervision:** Estrella I. Cruz, Ferdinand V. Salazar.

**Validation:** Ariza Minelle A. Aguila, Jennifer Ramos.

**Writing – original draft:** Ferdinand V. Salazar, Richard E. Paul.

**Writing – review & editing:** Ferdinand V. Salazar, Mary Vinessa Villaruel-Jagmis, Richard E. Paul.

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
