## [Decision Letter · Decision Letter 0]

3 Jan 2024

Dear Dr. Paul,

Thank you very much for submitting your manuscript "Current and lagged associations of meteorological variables and Aedes mosquito indices with dengue incidence in the Philippines" for consideration at PLOS Neglected Tropical Diseases. As with all papers reviewed by the journal, your manuscript was reviewed by members of the editorial board and by several independent reviewers. In light of the reviews (below this email), we would like to invite the resubmission of a significantly-revised version that takes into account the reviewers' comments. 

We apologize for the long duration of peer review, but are looking forward to a revised version of this manuscript. Please respond in detail to all reviewer comments and in particular, provide needed details on the methodologies used.

We cannot make any decision about publication until we have seen the revised manuscript and your response to the reviewers' comments. Your revised manuscript is also likely to be sent to reviewers for further evaluation.

Sincerely,

Alexandra K Heaney

Academic Editor

Audrey Lenhart

Section Editor

We apologize for the long duration of peer review, but are looking forward to a revised version of this manuscript. Please respond in detail to all reviewer comments and in particular, provide needed details on the methodologies used.

Reviewer's Responses to Questions

**Key Review Criteria Required for Acceptance?**

**Methods**

-Are the objectives of the study clearly articulated with a clear testable hypothesis stated?

-Is the study design appropriate to address the stated objectives?

-Is the population clearly described and appropriate for the hypothesis being tested?

-Is the sample size sufficient to ensure adequate power to address the hypothesis being tested?

-Were correct statistical analysis used to support conclusions?

-Are there concerns about ethical or regulatory requirements being met?

Reviewer #1: See attachment

Reviewer #2: The objectives and study setting are clear and are supported by the data collected. The sample size is sufficient, and I believe the correct statistical analyses were used (although there are many missing methodological details, making it difficult to evaluate this criteria). I have no concerns about the ethical requirements.

Reviewer #3: This study investigates the association between meteorological variables, Aedes mosquito indices and dengue incidence in six cities in the Philippines from October 2014 - December 2015. This is an interesting investigation into climate drivers of dengue, which capitalises on monthly entomological surveys capturing data on larvae and pupae in household water containers, as well as adult mosquitos inside the house.

I have the following questions and suggestions regarding Methods: 

1. How close were the meteorological stations to the selected cities? And how likely are they to be representative of climatic conditions in the cities?

2. It seems interesting that the proportions of immature Aedes sp were very different to adult Aedes mosquitos in the study sites (85.4% vs 50.5% for Aedes aegypti), do the authors have any intuition of why that may be?

3. It would be useful to understand how dengue incidence in the study year (2014-2015) compared to other dengue seasons - would it be considered an outbreak year? Similarly, it would be useful to understand the climate context of the study year chosen - were there any impacts from the 2014-2016 El Niño event? 

4. As authors comment in the manuscript, this is a short time series to characterise associations between meteorological variables and dengue, it would be useful to see a sensitivity analysis around the analyses focusing on climate and dengue with a longer time series (as I assume the entomological data is only available from 2014-2015)? 

5. I am curious as to why authors decided to use loglinear regression rather than Poisson or negative binomial regression for analyses concerning counts? Similarly, the arcsine transformation is not commonly used and it would be useful to understand why this was chosen.

6. In the methods section authors describe how the multivariable model was selected through a backwards selection process until a final adequate model with only significant variables was achieved. I’m interested as to whether other model adequacy criteria were used in model selection to avoid overfitting, e.g. WAIC? It would also be useful to understand whether this process was repeated for every outcome assessed.

7. In the methods section authors describe how highly correlated and colinear variables were excluded but final models seem to include several highly related variables (e.g. many models contain both cumulative rain and mean rain, while the model for Pupal index contains both minimum and maximum temperature). This would suggest that some of the final estimates of association between meteorological variables and the outcome may be confounded, could the authors add some more information on which colinear and correlated variables were excluded and which were kept in? 

8. The methods section also describes that “barangay was nested within city”, was this through a random effect? Model equations for the GAMs used (even if in the supplementary) would be very helpful in understanding the model structure. 

9. I assume that model results shown (tables of % variance explained and figures) were from the final multivariate model for each outcome but this is unclear and it would be helpful to add this information into the table and figure legends.

10. With regards to lagged meteorological variables, from Table 2 it seems that each meteorological variable was assessed at the same lag. This does not seem ideal as it’s likely that different meteorological variables will affect dengue incidence at differing time lags. It would be useful to see results from models tested with meteorological variables at different lags, particularly if an aim of this paper is to examine potentially useful models for early warning.

**Results**

-Does the analysis presented match the analysis plan?

-Are the results clearly and completely presented?

-Are the figures (Tables, Images) of sufficient quality for clarity?

Reviewer #1: See attachment

Reviewer #2: The results are described succinctly, but there are no corresponding figures, thus is it not possible for me to evaluate whether the descriptions match the data. The figures are provided at very low resolution.

Reviewer #3: 11. With regards to the figures showing associations between meteorological variables and mosquito / dengue outcomes, it would be preferable to show the non-linear association on the relative risk scale. This would help the viewer to interpret the magnitude of the non-linear effect of each meteorological variable on mosquito abundance / dengue risk. 

12. It is somewhat confusing to only see the results for some of the explanatory variables in the figures - it would be preferable to see all the results for one model as a figure, with panels for each association between the explanatory variable and the outcome, with similar figures for other outcomes in the Supplementary. This would help the reader interpret the relative effects of each variable on the outcome. It would also be useful to show confidence intervals in the results figures.

13. Figures 6 and 7 are somewhat confusing for the reader, it would be preferable to show each site as a separate line or as panels of the Figure rather than alternating between sites on the x-axes.

**Conclusions**

-Are the conclusions supported by the data presented?

-Are the limitations of analysis clearly described?

-Do the authors discuss how these data can be helpful to advance our understanding of the topic under study?

-Is public health relevance addressed?

Reviewer #1: See attachment

Reviewer #2: Again it is hard to evaluate whether the conclusions are supported by the data, as most of the relationships described are not presented in figures

Reviewer #3: 14. In Line 442 authors conclude that “the link between mosquito indices and dengue cases is non-significant”. It would be useful to see the model results for models with mosquito indices and dengue cases but without meteorological variables. From the current results, it seems this paper is showing that they don’t provide any additional explanation of variance when added to a meteorological model, which would be expected under a causal pathway in which meteorological variables affect dengue incidence through their effect on the vector population. It would be useful to add some discussion on the causal hypotheses examined here. Given that the study design in this paper is based on explaining variation in dengue incidence rather than predicting cases, it is also possible that the inclusion of mosquito indices could still be useful in a predictive early warning model and I would recommend amending this conclusion.

**Editorial and Data Presentation Modifications?**

Reviewer #1: (No Response)

Reviewer #2: Major revisions

Reviewer #3: I think there was an issue with rendering in Figure 1 making the text difficult to read, some of the text on map inserts is illegible.

**Summary and General Comments**

Reviewer #1: See attachment

Reviewer #2: (The text below is copied from the review document I have uploaded outlining my summary, major, and minor comments).

Overall description:

This study investigates the impact of meteorological variables and mosquito indices on dengue incidence, to support the development of early warning and response systems. While the general findings are not particularly novel (as there are now countless studies investigating climate and weather impacts on mosquito-borne disease transmission), these relationships are likely place- and context-specific, thus conducting this investigation in key cities in the Philippines is of merit. Further, the authors found a strong relationship between recent weather conditions, mosquito abundances and dengue cases which could be useful for developing local early warning systems. However, it is very challenging to rigorously evaluate the specific findings presented here as there are many key methodological details and figures missing (outlined further below), 

Major comments:

- I would like to evaluate whether the qualitative descriptions provided in the discussion match the data, however the relationships between climate variables and mosquito indices are not depicted anywhere (only the statistical significance is reported in the tables. Further, it is hard to follow the results given the high number of models and specifications used here. Namely, is there a separate model regressing each of the mosquito indices against meteorological variables (6 models + an additional 24 models for the 1-4 week lags). Within these 30 models, are there separate univariate and multivariate models? Is there then another 6 models (+ lagged versions) using each of these mosquito indices as the univariate predictor of dengue cases? And then one additional model using all meteorological variables and mosquito indices as predictors of dengue? The results are described succinctly, which is helpful, but it is difficult to understand which model each result is derived form. 

- It is surprising that using 1-4-week lagged temperature variables explained less variation than the non-lagged temperature variables (as it is very often the case that temperature and rainfall in the prior month have a larger impact on mosquito abundance than the equivalent variables in the current month). More methodological details are critical for making sense of these results. Namely, are the dengue cases used here attributed to the month in which it was reported? If so, does it not likely reflect transmission that occurred in the weeks prior? Further, were the models re-fit using these lagged variables? Or were the same variables as identified originally used and then various lags applied? I think the former would be more appropriate, as you wouldn’t necessarily expect the same variable to be important across time scales (e.g., precipitation may matter more in the 3-4 weeks prior as it would impact breeding habitat, but max temperature may matter more in the 1-2 weeks prior). This also applies to the model regressing dengue cases against meteorological variables.

- I might expect slightly different meteorological variables to be important for Aedes aegypti vs albopictus (particularly as albopictus has lower heat tolerance). It would be helpful to know if models generated for Ae. aegypti matched those generated using the two species combined. 

- It would be helpful to know if any environmental data was collected from inside the homes themselves. I understand it is not possible to back and collect this information (and it would no longer be relevant), but prior work has shown there can be substantial differences in temperature in indoor and weather-station recorded temperatures (e.g., Pena-Garcia et al. 2023) 

Minor comments:

• Line 65 – citation needed for these case counts in the Philippines. 

• line 66 – typo (extra word ‘incidence’)

• Line 75 – could also add Ryan et al. 2019 (Plos NTD) 

• Line 80 – the overall impacts of temperature on mosquito life history traits is also nicely summarized in Mordecai et al. 2020 (Ecology Letters)

• Line 87 – suggest either defining / describing what Integrated Vector Management practices and Communication for Behavioral Impact approaches are, or removing these details 

• Line 92 – perhaps worth noting that a description of these indices (‘House, Container, Breteau and Pupal’) can be found in the methods 

• Line 145 – could you add further details about surveillance of these water-holding containers. Namely, how far from the house did your survey extend? What types

---

## [Decision Letter · Decision Letter 1]

29 Apr 2024

Dear Dr. Paul,

Thank you very much for submitting your manuscript "Current and lagged associations of meteorological variables and Aedes mosquito indices with dengue incidence in the Philippines" for consideration at PLOS Neglected Tropical Diseases. As with all papers reviewed by the journal, your manuscript was reviewed by members of the editorial board and by several independent reviewers. In light of the reviews (below this email), we would like to invite the resubmission of a significantly-revised version that takes into account the reviewers' comments. 

We apologize for the delay in getting back to you. It has been difficult to reach reviewers. Thank you for your first round of revisions, they have significantly strengthened your manuscript. In this next round, please respond in particular to the continued concerns of Reviewer 2.

We cannot make any decision about publication until we have seen the revised manuscript and your response to the reviewers' comments. Your revised manuscript is also likely to be sent to reviewers for further evaluation.

Sincerely,

Alexandra K Heaney

Academic Editor

Audrey Lenhart

Section Editor

We apologize for the delay in getting back to you. It has been difficult to reach reviewers. Thank you for your first round of revisions, they have significantly strengthened your manuscript. In this next round, please respond in particular to the continued concerns of Reviewer 2.

Reviewer's Responses to Questions

**Key Review Criteria Required for Acceptance?**

**Methods**

-Are the objectives of the study clearly articulated with a clear testable hypothesis stated?

-Is the study design appropriate to address the stated objectives?

-Is the population clearly described and appropriate for the hypothesis being tested?

-Is the sample size sufficient to ensure adequate power to address the hypothesis being tested?

-Were correct statistical analysis used to support conclusions?

-Are there concerns about ethical or regulatory requirements being met?

Reviewer #1: yes

Reviewer #2: The clarify of the methods has been much improved since the original version, but some pertinent details remain missing (As described in attachment)

**Results**

-Does the analysis presented match the analysis plan?

-Are the results clearly and completely presented?

-Are the figures (Tables, Images) of sufficient quality for clarity?

Reviewer #1: yes

Reviewer #2: Additional figures have been provided, as recommended. However, they remain very low resolution (perhaps just for me).

**Conclusions**

-Are the conclusions supported by the data presented?

-Are the limitations of analysis clearly described?

-Do the authors discuss how these data can be helpful to advance our understanding of the topic under study?

-Is public health relevance addressed?

Reviewer #1: yes

Reviewer #2: The limitation are now more sufficiently addressed. However, given the authors assertion that this study is an attempt to develop an early warning system, rather than a climate-dengue study, the framing of the manuscript seems off (I discuss more in the attachment).

**Editorial and Data Presentation Modifications?**

Reviewer #1: none

Reviewer #2: (No Response)

**Summary and General Comments**

Reviewer #1: The reviews are essential and have improved the manuscript a lot. I don't have any further comments.

Reviewer #2: The topic remains highly topical and important for public health. I think this manuscript could make a contribution if it is re-framed to focus more heavily on what factors predict dengue cases specifically, an evaluation of the warning system, and how it may be applied.

PLOS authors have the option to publish the peer review history of their article (what does this mean?). If published, this will include your full peer review and any attached files.

Reviewer #1: No

Reviewer #2: No
---

## [Decision Letter · Decision Letter 2]

27 Jun 2024

Dear Dr. Paul,

We are pleased to inform you that your manuscript 'Current and lagged associations of meteorological variables and Aedes mosquito indices with dengue incidence in the Philippines' has been provisionally accepted for publication in PLOS Neglected Tropical Diseases.

Best regards,

Alexandra K Heaney

Academic Editor

Audrey Lenhart

Section Editor

Reviewer's Responses to Questions

**Key Review Criteria Required for Acceptance?**

**Methods**

-Are the objectives of the study clearly articulated with a clear testable hypothesis stated?

-Is the study design appropriate to address the stated objectives?

-Is the population clearly described and appropriate for the hypothesis being tested?

-Is the sample size sufficient to ensure adequate power to address the hypothesis being tested?

-Were correct statistical analysis used to support conclusions?

-Are there concerns about ethical or regulatory requirements being met?

Reviewer #2: Regarding the dengue case data — the authors have still not addressed my original comment requesting additional information about the dengue case data. Namely - are dengue cases ascribed to the month they were reported? How was this case data collected and from where did the authors access it?

**Results**

-Does the analysis presented match the analysis plan?

-Are the results clearly and completely presented?

-Are the figures (Tables, Images) of sufficient quality for clarity?

Reviewer #2: I believe there is some error occurring in the submission system whereby the figures I see remain very low resolution, despite the authors confirmation that they have uploaded suitable versions.

**Conclusions**

-Are the conclusions supported by the data presented?

-Are the limitations of analysis clearly described?

-Do the authors discuss how these data can be helpful to advance our understanding of the topic under study?

-Is public health relevance addressed?

Reviewer #2: I appreciate the authors efforts to shift the framing of the paper to better emphasize their goal of assessing the added value of measuring mosquito indicates for early-warning systems.

**Editorial and Data Presentation Modifications?**

Reviewer #2: None

**Summary and General Comments**

Reviewer #2: No further comments. I appreciate the authors attentiveness to the reviewer's concerns

PLOS authors have the option to publish the peer review history of their article (what does this mean?). If published, this will include your full peer review and any attached files.

Reviewer #2: No

---

## [Editor Report · Acceptance letter]

10 Jul 2024

Dear Dr. Paul,

We are delighted to inform you that your manuscript, "Current and lagged associations of meteorological variables and Aedes mosquito indices with dengue incidence in the Philippines," has been formally accepted for publication in PLOS Neglected Tropical Diseases.

Best regards,

Shaden Kamhawi

co-Editor-in-Chief

Paul Brindley

co-Editor-in-Chief
